# Conducting Polymer-Based Nanohybrids for Fuel Cell Application

**DOI:** 10.3390/polym12122993

**Published:** 2020-12-15

**Authors:** Srabanti Ghosh, Suparna Das, Marta E. G. Mosquera

**Affiliations:** 1Department of Organic and Inorganic Chemistry, Universidad de Alcala (UAH), 28805 Alcalá de Henares, Madrid, Spain; martaeg.mosquera@uah.es; 2Department of Chemical and Biomolecular Engineering, University of California, Irvine, CA 92697, USA; suparnad@uci.edu

**Keywords:** fuel cell, conducting polymer, nanohybrids, catalysts support, electrooxidation, functionalization, anode catalysts, cathode catalysts, maximum power density

## Abstract

Carbon materials such as carbon graphitic structures, carbon nanotubes, and graphene nanosheets are extensively used as supports for electrocatalysts in fuel cells. Alternatively, conducting polymers displayed ultrahigh electrical conductivity and high chemical stability havegenerated an intense research interest as catalysts support for polymer electrolyte membrane fuel cells (PEMFCs) as well as microbial fuel cells (MFCs). Moreover, metal or metal oxides catalysts can be immobilized on the pure polymer or the functionalized polymer surface to generate conducting polymer-based nanohybrids (CPNHs) with improved catalytic performance and stability. Metal oxides generally have large surface area and/or porous structures and showed unique synergistic effects with CPs. Therefore, a stable, environmentally friendly bio/electro-catalyst can be obtained with CPNHs along with better catalytic activity and enhanced electron-transfer rate. The mass activity of Pd/polypyrrole (PPy) CPNHs as an anode material for ethanol oxidation is 7.5 and 78 times higher than that of commercial Pd/C and bulk Pd/PPy. The Pd rich multimetallic alloys incorporated on PPy nanofibers exhibited an excellent electrocatalytic activity which is approximately 5.5 times higher than monometallic counter parts. Similarly, binary and ternary Pt-rich electrocatalysts demonstrated superior catalytic activity for the methanol oxidation, and the catalytic activity of Pt_24_Pd_26_Au_50_/PPy significantly improved up to 12.5 A per mg Pt, which is approximately15 times higher than commercial Pt/C (0.85 A per mg Pt). The recent progress on CPNH materials as anode/cathode and membranes for fuel cell has been systematically reviewed, with detailed understandings into the characteristics, modifications, and performances of the electrode materials.

## 1. Introduction

Fuel cells have appeared as potential energy conversion devices which operate in the presence of the hydrogen or hydrogen-rich fuels with low CO_2_ emissions [1,2,3]. Among these, proton exchange membrane fuel cells (PEMFC) have gained more attention due to their superior properties such as high open-circuit potential, high energy conversion efficiency, limited fuel crossover effects, and efficient electrooxidation of organic molecules such as ethanol (DEFC), methanol (DMFC), formic acid (DFAFC), etc. at low-temperature [4,5,6,7].

On the other hand, microbial fuel cells (MFCs) provide the possibility of transforming organic waste directly into electricity through microbially catalyzed anodic and microbial/enzymatic/abiotic cathodic electrochemical reactions with solid electrodes [8,9].The main reasons which hinders the practical application of the MFC are the lower power output and high cost Pt-based catalysts. Therefore, various carbonaceous, metallic nanoparticles (NPs) as anode materials and carbonaceous, platinum-group metal and platinum-group-metal-free materials as cathode catalysts are used to improve the performance of the MFC along with lowering the cost [10,11,12].

Despite significant progress, high material cost and limited availability of Pt together with intrinsic sluggish kinetics of anodes limit the commercial use of FCs [13]. Moreover, stability and efficiency of the fuel cell has been associated with the optimized three-phase interface that is composed of an ion-conducting electrolyte membrane (ionomer) attached to the catalyst surface [14,15,16]. Initially, hydrogen gas molecules react with the solid phase of the anode, and protons are accessible to the electrolyte phase and then react with O_2_ in the solid–electrolyte interface. Thus, the development of stable, low-cost electrocatalysts and polymer membranes, and understanding the effectiveness of catalytic activity of small organic molecules are crucial to designing large-scale and cost-effective fuel cells.

Up until now, it has been demonstrated that carbon supports such as carbon black, carbon nanotubes (CNTs), carbon fibers, mesoporous carbon, graphitic carbons, graphene etc. with suitable surface properties and functional groups can be used for metal NP deposition to fabricate efficient catalysts FCs [17,18,19,20,21,22,23,24,25,26]. However, significantly lower interactions between hydrophobic surfaces of carbon supports with the catalyst and corrosion of supporting materials under experimental conditions hinder high catalytic activity and durability, which remain a significant challenge [27,28]. Moreover, the efficiency and power density of alcohol FCs may be strongly affected due to the fuel crossover through the traditionally used Nafion membrane [29]. Alternatively, conducting polymers (CP) such as polypyrrole (PPy), polyaniline (PANI), poly-3-methyl thiophene (PMT), poly(3,4-ethylenedioxythiophene) (PEDOT), etc. have been successfully used as supporting material for electrocatalysts in fuel cell applications [30,31,32,33,34]. Conducting polymers have exhibited promising applications as catalyst supports due to unique conjugated structures with high electrical conductivity, exceptional chemical stability, facile fabrication, and low cost. Thus, conducting polymer-based nanohybrids (CPNHs) with a series of metal or metal oxides deposited on the polymer surface have been employed for electrooxidation of small molecules such as hydrogen, methanol, and formic acid [35,36]. Recently, multidirectional efforts have been devoted to fabricate electrode materials via depositing the nanostructured metal catalysts on conducting polymer surfaces directly or on modified polymer nanostructures [37,38].

The present review provides an overview on the use of conducting polymer-based nanohybrids as active electrode materials or catalyst supports for PEM and MFC applications. The recent developments of different methods to modify the polymer-based hybrid nanostructures for both anode and cathode electrocatalysts with enhanced electrocatalytic performances are described. These results provide fundamental insight into how CPNH materials can modify the performance of electrocatalysts which have potential as advanced electrode materials for fuel cell applications.

## 2. CPNH-Based Electrode Materials

Conducting polymer-supported nanomaterials represent a unique class of hybrid electrocatalysts for fuel cells applications that synergize the beneficial properties of both nanomaterials and conducting polymers [35]. The higher electronic conductivity of CPs compared to that of the conventional carbon-based catalyst support this higher attention during past two decades. Additionally, the higher surface area and unique synergistic effects with the metal NPs/metal oxide play an important role to improve the electrocatalytic activity of the catalysts and to stabilize fuel cell performance [39,40]. A significant effort has been made in the fabrication of Pt-free catalysts deposited on CPs, although loading of low-Pt nanomaterials on polymers is also used due to their high catalytic performance. Various synthetic strategies have been developed successfully by several research groups for conducting polymer-supported hybrid catalysts, including metal NPs, multimetallic NPs, and metal oxides which can be used as anode, cathode, and electrolyte membrane, as shown in Figure 1.

### 2.1. CPNH-Based Electrode for Alcohol Fuel Cells

Conducting polymer-supported hybrid electrocatalysts have been widely used for the oxidation of small organic molecules and oxygen reduction, which are relevant to fuel cells. In principle, direct alcohol fuel cells (DAFCs) convert the chemical energy stored in the alcohol molecules into electricity through electrooxidation at the anode in the presence of catalysts (Figure 2). For example, the oxidation of methanol in alkaline or acid media is a six-electron oxidation process that forms CO_2_ (Equations (1) and (2)):CH_3_OH + 6 OH^−^→ CO_2_ + 5 H_2_O + 6 e^−^ (at anode)(1)
CH_3_OH + H_2_O → CO_2_ + 6H^+^ + 6e^−^(2)

Another commonly used fuel, ethanol involves an alkaline oxidation reaction (EOR), which generates carbon intermediates as well as acetaldehyde (CH_3_CHO), acetic acid (CH_3_COOH), and 12 e^−^ (Equations (3) and (4)):CH_3_CH_2_OH +12 OH^−^→ 2 CO_2_ + 9 H_2_O +12 e^−^ (at anode)(3)
3 O_2_ + 6 H_2_O +12 e^−^→12 OH^−^ (at cathode)(4)

Up to now, Pt-based catalysts have been utilized as an effective electrocatalyst for alcohol fuel cells, and a significant effort has been made to lower the cost of such catalysts through lowering Pt loadings [41]. Moreover, low-cost Pd also demonstrated efficient alcohol oxidation in alkaline media with high catalytic activity and anti-CO-poisoning [42,43]. A series of conducting polymers such as PPy, PANI, PEDOT, etc. (as shown in Figure 1) with large surface areas, high conductivity, and shortened pathways for charge/mass transport have been successfully used as catalyst supports for fuel cell applications.

On the other hand, in microbial fuel cells, microorganisms employed as catalysts to generate electric current from biodegradable organic and inorganic compounds may be useful for a wide range of applications including electricity generation, bio-hydrogen production, waste water treatment, medical devices, and other electrochemical devices [44,45,46]. MFC is mainly composed of an anode; a cathode (example, graphite, graphite felt, carbon paper, carbon cloth etc.); a polymer electrolyte membrane, such as nafion, ultrex, polyethylene, poly(styrene-co-divinylbenzene), etc. substrates; and electrode catalysts (Figure 2b). A wide range of metal and metal-based catalysts, like Pt, Pt black, and MnO_2_ have been used as catalysts for the electrodes, mainly for cathodes, while microorganisms themselves act as catalysts in the anode. Nowadays, to speed up the anode reaction, nontoxic graphene-based catalysts have also been used. Electrons flow from the anode to the cathode through an external electrical connection. Anodic oxidation of the substrate, including carbohydrate, proteins, volatile acids, cellulose, or waste water (for example, acetate) in the presence of microbes produced electrons and protons as well as carbon dioxide (Equations (5) and (6)).
CH_3_COO^−^ +2 H_2_O → 2 CO_2_ + 7 H^+^ + 8 e^−^ (at anode)(5)
O_2_ + 4H^+^ + 4 e^−^→ 2 H_2_O (at cathode)(6)

#### 2.1.1. Nanohybrid Electrode Materials Using CPs

The metal NP formation occurring with synthesis of the conducting polymer via a one-pot method is highly desirable since the inorganic/organic hybrid materials synergize the properties of both components as depicted in Figure 3a and lead to the development of high-performance catalysts [34,35] as prepared catalysts are characterized by using common techniques like electron microscopy, optical spectroscopy, scattering, and cyclic voltammetry (CV) in combination with electrochemical impedance spectroscopy. For example, conducting polymer nanofibers are formed at the interface of a lamellar structure-based surfactant-mediated soft template in the presence of a chemical oxidant (ammonium persulfate) and Pd metal salts [47]. Figure 3b shows the transmission electron microscopy (TEM) image, which demonstrates the formation of well-dispersed Pd NPs of 4–7 nm deposited on the PPy nanofiber having diameter ca. 28 nm, and the length is more than 2 μm, as determined by an in situ technique. The presence of Pd NPs with face-centered cubic (fcc) Pd on PPy nanofibers are confirmed by X-ray powder diffraction, and the composition of Pd/PPy catalysts consists of Pd signal as well as Cl, C, N, and O from PPy observed from X-ray photoelectron spectroscopy. Comparing the peaks of Fourier transformed infrared spectroscopy (FTIR) of PPy and Pd/PPy nanohybrids (NHs) suggests that shifting of bands of polypyrrole occurred due to a strong interaction between Pd NPs and that polymer matrix and PPy nanofibers may act as a capping agent for Pd NPs. Moreover, N_2_ adsorption–desorption measurement demonstrated mesoporous structures of Pd/PPy, which promote easy access of electrolytes on the surface useful for catalytic applications. Electrochemical measurements using CV and chronoamperometry (CA) indicate that the Pd NP-based polymer nanohybrids show high electrocatalytic activity for ethanol electrooxidation. The Pd/PPy NHs have been used as anode catalysts for the electrocatalytic ethanol oxidation reaction (EOR),where superposition of the first cyclic voltammogram (black solid line curve) and the 100th cycle (red solid line curve) of Pd/PPy NHs run in 1 M KOH containing 1M EtOH (Figure 3c). For comparison, Pd/PEDOT NHs and Pd/PANI NHs were also prepared and showed similar voltametric features as the onesfor ethanol oxidation reaction. However, the specific catalytic activities of the catalysts follow the order Pd/PPy (13.08 mA cm^−2^) > Pd/PANI 8.18 mA cm^−2^) > Pd/PEDOT (5.08 mA cm^−2^). The high catalytic activity is attributed to the strong interaction between Pd and PPy polymer nanofibers compared to other polymers.

Pandey et al. [48] showed excellent catalytic activity toward ethanol electro-oxidation in an alkaline medium of a Pd-PEDOT film prepared by the one pot electrochemical method through the formation of Pd NPs by the dissolution of Pd anode and simultaneous oxidative polymerization of EDOT monomer and the subsequent electrophoretic deposition on the gold substrate as a thin film. Further, the electro deposition of the Pd-PEDOT film in galvanostatic mode with different current values was followed to evaluate the effect of the current on the nature of the coating. This one pot synthetic procedure is simple and can be performed at room temperature without the need for templates and surfactants. In another method, polymers are initially prepared by soft template synthesis method and then monometallic, bimetallic, or trimetallic NPs are deposited on a polymer surface by a chemical reducing agent or by UV radiation or gamma irradiation without using any chemical reducing agent, as shown in Figure 3d. PPy nanofibers were synthesized using soft oxidative templates, and then, a unique Pd-branched structure was formed on the polymer surface by the photo-reduction method as shown in the TEM image (Figure 3e) [49]. The polymer-supported Pd electrocatalysts demonstrated an excellent EOR performance in terms of high current density for the Pd branched structure with the onset potential (E_onset_) shifted to a more negative potential compared to commercial Pd/C, suggesting enhancement in the kinetics of ethanol oxidation (Figure 3f).

Ghosh and coworkers [50] developed a unique technique using radiation-induced synthesis of the multi-metallic nanoalloys on the conducting polymer nanofibers as new generation metal/polymer hybrid materials for electrocatalytic application in direct ethanol fuel cells. Different combinations of metal nanoalloys (M= Pd–Pt, Pd–Au, and Pd–Pt–Au) with tunable compositions can be prepared, and high current density as well as long-term stability was observed for the ethanol oxidation compared with the commercial Pd/C catalysts. Thermogravimetric analysis (TGA) showed a higher decomposition temperature with high residual mass of PPy, which indicates metal loading about approximately 19–48%, which is consistent with the inductively coupled plasma atomic-emission spectroscopy (ICP-AES) technique. Figure 3g displays trimetallic Pd_30_Pt_29_Au_41_NPs dispersed on the PPy polymer surfaces with average particle sizes of approximately 8 nm. Pd_30_Pt_29_Au_41_/PPy nanohybrids display excellent electrocatalytic activities which are approximately 17 and 9 times higher than Pd/PPy and Pd/C, respectively. The CA measurement at constant potentials for ethanol electro-oxidation of the Pd_30_Pt_29_Au_41_/PPy electrode exhibited higher limiting as well as initial current, suggesting superior stability than the Pd/PPy catalysts (Figure 3h). A similar trend was followed in the stability of the electrodes up to 1000 cycles, and the current densities of the Pd_30_Pt_29_Au_41_/PPy and Pd_54_Au_46_/PPy electrodes still remained at approximately 93.4% and 97% compared to Pd/PPy (53%) and 100% decay for commercial Pd/C, as shown in Figure 3i.

Additionally, multilayered PtPd/PPy/PtPd [51] and Pd/PANI/Pd [52] were also developed as highly active electrocatalysts for the oxidation of a series of small organic molecules such as methanol, ethanol, and formic acid, which allow for rapid transport of electroactive species and high surface area. Thus, highly dispersed, ultrasmall metal NPs can be grown directly on the surface of the conducting polymer surface without using any additional linker, and in fact, such low metal content loading catalysts with high performance are highly desirable for fuel cell applications (Table 1).

Moreover, Pd/PPy showed high catalytic performance towards the oxygen reduction reaction (ORR) in 0.5 M H_2_SO_4_ with positively shifted half-wave potential and four-electron reduction generating water as the main product and the fraction of H_2_O_2_ [53].They proposed that oxygen migration may happen through the PPy matrix and may further diffuse through the solution to reach the Pd NPs deposited on the polymer surface. The Au‒PANI hybrid electrode showed excellent electrocatalytic activity toward the electrochemical O_2_ reduction, which may associate with the fast charge transfer kinetics and high selectivity for O_2_ reduction to water (OH^−^) [54].Very recently, Verma et al. [55] fabricated Au-V_2_O_5_/Pin hybrid electrodes through insitu chemical polymerization of an indole monomer in the presence of nano V_2_O_5_ dispersion using HAuCl_4_ as an oxidant for ORR.

A series of Pt NP-supported conducting polymer nanohybrid-based electrocatalysts such as platinum NPs deposited on PEDOT [56], PPy containing catalysts [57], platinum catalyst-supported PANI nanotubules [58], Pt NP-loaded PANI hollow tubes [59], and Pt nanoclusters embedded on poly(*N*-acetylaniline) nanorods [60] have been employed for electrocatalytic oxidation of methanol and ethanol. Notably, two-dimensional polyaniline nanosheets deposited with high areal density Pt nanocrystals of 2.7 nm showed good electrocatalytic performance for methanol oxidation, both in activity and stability, by Kim et al. [61]. PPy nanofibers with well-dispersed Pt NPs exhibited a higher catalytic activity (14.1 mA cm^−2^), significantly higher than that of commercial carbon black powder supports (4.6 mA cm^−2^) by Viswanathan and coworkers [62]. Guo et al. [63] reported PANI nanofiber-supported high-density Pt NPs by in situ chemical polymerization, which showed the current density to be 2.99 times higher than that of the Pt/C catalyst for methanol oxidation. A unique core–shell superstructure consisting of Pt nanocube assemblies with PANI reported as a highly efficient methanol oxidation reaction (MOR) electrocatalyst with specific activity of Pt NCAMs@PANI is 0.85 mAcm^−2^, which is 2.58 times higher than that of commercial Pt/C (0.33 mAcm^−2^) [64]. Moreover, earth-abundant Ni metal catalysts supported with polyaniline (PAni) and partially sulfonated PAni (SPAni) have been applied as an alternative to the Pt metal catalyst for oxidation of methanol by Das et al. [65]. Notably, a high current density of 2.15 mA cm^−^^2^ at +0.2 V was obtained using Ni/SPAni as anode catalysts for MOR due tobetter dispersion, smaller particle size, and higher utilization of Ni NPs on the SPAni matrix compared to that on the carbon-supported materials.

In addition, in conducting polymer-containing bimetallic and trimetallic hybrid catalysts, such as Pt–Ru/PPy, Pt–Fe/PPy, and Pt–Co/PPy, Pt–Pd/PPy, Pt–Pd–Au have also been explored for methanol electrooxidation. Remarkably, poly(pyrrole) hollow spherical nanocapsules have been used as an efficient support matrix for PtRuNPs, which are expected to have three-dimensional access to the electroactive species and to be potential catalysts for fuel cell [66]. Further, improved kinetics for methanol oxidation with onset potential was reduced by 220 mV using poly(o-phenylenediamine) (PoPD)-Pt–Ru nanohybrid electrocatalysts compared to the (PoPD)-Pt electrode by Gajendran et al. [67]. In another example, Pt/Cu bimetallic nanomaterials deposited on poly 3,4-ethylenedioxythiophene with a relatively low Pt loading was tested as an anode catalyst for direct methanol fuel cells [68]. Zhao et al. [69] fabricated a bimetallic Pt–Fe/PPy–carbon catalyst which showed improved catalytic activity with reduced onset potential and 1.5 times higher anodic peak current density towards methanol oxidation compared to a commercial Pt/C catalyst. Highly dispersed bimetallic PtPd and trimetallic PtPdAu nanoalloys were deposited on PPy nanofibers by the radiolytic method and showed significantly improved catalytic activity and durability for methanol oxidation compared to Pt/C and Pt/PPy electrodes [70]. The current density of Pt_66_Pd_34_/PPy NHs is approximately 4.8 times higher than Pt/PPy NHs, which suggests that bimetallic catalysts demonstrated significant catalytic activity towards MOR, while Pt_24_Pd_26_Au_50_/PPy NHs displayed high performance towards methanol oxidation, which may be associated with controlling the poisoning effect by alloying of Pt with Pd, and the presence of Au enhances the reactivity.

#### 2.1.2. Nanohybrid Electrode Materials Using Modified CPs as Support

Traditional carbon-based supports such as carbon black, carbon nanotubes (CNTs), carbon fibers, mesoporous carbon, graphene nanosheets, etc. have high hydrophobic surfaces, which make them suitable for fuel cell applications. Metal catalysts leaching from such carbon supports may be overcome through functionalization with the polymers that ideally can be used as supporting material for electrooxidation of alcohol molecules [71]. The following table (Table 2) summarizes the latest publications on functionalized conducting polymer-based supports in electrocatalysts. For example, 3,4-polyethylenedioxythiophene-coated carbon paper used as supporting materials for Pt NPs (Figure 4a) displayed efficient electrooxidation of methanol and superior electrochemical activity due to faster kinetics of MOR in the presence of a conjugated polymer compared to the carbon support alone [72]. In another report, Dash et al. [73] explored the role of PEDOT-modified carbon paper-deposited Pd nanodendritic hybrid electrodes for the electrooxidation of a series of alcohols. Initially, PEDOT was electrochemically deposited on carbon paper, where the morphology of the polymer layer depends strongly on the potential of deposition and a globular type of morphology obtained at 0.90 V while 0.07 C cm^−2^ charge was used for a fixed quantity. Then, well-dispersed Pt NPs were coated onto PEDOT by potentiostatic electrodeposition in 0.1 M H_2_SO_4_ and at a potential of 0.10 V. Electrocatalytic activity of the Pt-PEDOT/C electrode showed superior performance for methanol oxidation compared to the Pt/C electrode. Joiceet al. [74] showed superior electrochemical activity of nanocactus Pt-PANI/ carbon fiber paper (CFP) electrode for toluene oxidation, which is higher than that of the Pt/CFP electrode. Ghosh et al. [75] used Nafion-based ion-exchange membrane-modified poly(diphenylbutadyine) (PDPB) nanofiber-supported Pd nanoplate-based nanohybrid electrodes for efficient ethanol oxidation, which associated with a strong interaction between the Pd nanostructures and the polymer support. Research findings revealed that a combination of Vulcan XC-72 and PANI doped with trifluoromethane sulfonic acid as a support for the Pt/C-PANI hybrid catalyst displayed considerable high catalytic activity and stability toward methanol oxidation [76]. Kuoet al. [77] also proposed conjugated polymer PEDOT modified with poly(styrene sulfonic acid) (PSS) as a support and Pt NPs deposited on modified polymer support for methanol oxidation. Similarly, Liu et al. [78] used a nanofibrous network of polyaniline–poly(styrene sulfonic acid) deposited with Pt–Ru for enhanced electrocatalytic activity toward methanol oxidation with a significantly less poisoning effect of CO. Recently, Ye et al. [79] introduced a combined system with a strong electrostatic attraction between PANI and a proton (perfluorosulfonic acid (PFSA), which improved the long-term stability of Pt electrocatalysts for oxygen reduction reaction compared to commercial porous carbon nanosphere-supported Pt catalysts. Pt NPs deposited on polypyrrole-carbon composites displayed superior catalytic activity both as anodes and cathodes for borohydride oxidation reaction in an alkaline medium and for hydrogen peroxide reduction reaction in an acidic medium [80].

Moreover, conducting polymers modified with metal oxides, carbon nanotubes, graphene, etc. have also been used as supporting materials for metal catalysts. Kuo et al. [81] initially prepared platinum NPs and hydrous molybdenum oxide (Pt/H_x_MoO_3_) and then electrodeposited then onto poly(3,4-ethylenedioxythiophene)-poly(styrene sulfonic acid) (PEDOT-PSS) film, which showed high electrocatalytic activity for methanol oxidation. The same research group used a composite consisting of ruthenium oxide particles into Pt and polyaniline-poly(acrylic acid-co-maleic acid) electrodes and tested for methanol oxidation [82]. It is noted that that PANI-PAMA-Pt-RuO_2_ electrode showed the best electrocatalytic activity and stability compared to the PANI-PAMA-Pt electrode. Recently, multilayered Pt/CeO_2_/PANI and ZnO/Pt/CeO_2_/PANI hybrid hollow nanorod arrays as illustrated in Figure 4b displayed higher electrocatalytic activity for methanol oxidation, as reported by Xu et al. [83]. They proposed that the presence of CeO_2_ can effectively release the poisoning of carbonaceous species and allowed dispersion of metal catalysts, facile electron transport, and electron delocalization due to the presence of the conducting polymer. Since metal oxides have low cost, high electrochemical stability, accessibility of surface hydroxyl groups, and strong interactions with metal nanoparticles, they significantly improve the catalytic performance of polymer-based nanohybrid electrodes [84]. Functionalized single-walled carbon nanotubes have been employed as support to improve catalytic performance and stability of the electrocatalysts for fuel cell applications [85,86]. Highly dispersed Pt–Ru/Polypyrrole–CNT and Pt–Ru/Polythiophene/CNT showed high catalytic activity for methanol oxidation and ethylene glycol oxidation respectively, having high electrochemically accessible surface areas, high electronic conductivity and facile charge-transfer [87,88]. Reddy et al. [89] developed a MWCNT supported Co-PPy composites as electrode material for oxygen reduction reaction without a noticeable loss in the performance over long operating times. In order to improve the catalytic activity, metal oxides modified poly(3,4-ethylenedioxythiophene)-carbon nanotubes composites have been used as support for Pt NPs in methanol electrooxidation by Wei et al. [90]. Fard et al. [91] fabricated Pd nanoflowers on a PPy modified MWCNTs support which exhibits the enhanced electrocatalytic activity with the mass activity of Pd NFs/PPY@MWCNTs (725 mA mg^−1^) being 8.09 times higher than that of the Pd NFs catalyst (89.6 mA mg^−1^). Research finding revealed that nanohybrids based onMWCNTs and conducting polymer form a “fiber in a jacket” organization and nanoparticles may immobilize in the polymer layer, which in turn enhance the electronic and protonic conductance, thermal stability, hydrophilicity, surface area of metal catalysts. This can be expected to influence the catalytic performance.

Graphene-modified conducting polymer-based composites were widely used as supporting materials for metal catalysts in fuel cell application (Figure 4d) [92]. The synergistic effect of metal nanoparticles and polymer plays a crucial role in enhancing the electrocatalytic performance of the NHs. In general, metal NPs are uniformly dispersed on polymer surfaces and uniformly wrapped by graphene sheets, which may accelerate the charge transfer between the nanohybrids and the electrolyte. Vertically oriented palladium NPs anchoring on polyaniline-reduced graphene oxide hybrid nanosheets have been tested as anode materials for methanol and ethanol oxidation [93]. Yue et al. [94] examined that Pd–PEDOT/graphene NHs displayed high electrocatalytic activity and enhanced the CO-antipoisoning ability and catalytic stability for ethanol oxidation. They proposed that the superior electrocatalytic performance of Pd–PEDOT/graphene may originate from the uniform dispersion of small Pd NPs (approximately 3.6 nm) on PEDOT nanospheres and on graphene nanosheets, which may provide a higher electroactive surface area and the formation of a 3D conducting frame, allowing facile charge transfer between nanohybrid catalysts and the electrolytes. Choe et al. [95] reported that PEDOT-functionalized graphene with palladium NPs showed remarkable electrocatalytic activity for ORR and stability compared to commercial Pt/C. Recently, Eswaran et al. [96] examined that graphitic carbon nitride/polyaniline/palladium NP composites can be utilized as stable electrode methanol oxidation compared to commercial palladium-loaded carbon black.

## 3. CPNH-Based Electrode for Microbial Fuel Cells

### 3.1. CPNH-Based Anode for MFCs

Performance of the MFC mainly depends on the functioning of the electrode materials, reactor configuration, and seed culture. At the interface of the anode catalyst layer and the microbe layer, the generated extracellular electron transference is considered a crucial factor for MFC performance. Therefore, a variety of electrodes such as graphite plate; titanium plate; gold sheets; or carbon materials like brush, cloth, paper, mesh etc. was employed as anodes of MFCs to improve its performance. However, several barriers, for example, non-porous structure, limited surface area, and poor contact between microbes and electrodes, limit the overall performance of the MFCs [97,98]. Recently, non-precious metal oxide catalysts have attracted great interest because of their high catalytic activity, lower cost, and biocompatibility. Again, agglomeration and dissolution of this metal oxide results in deterioration of the electrocatalytic activity of catalysts, which causes a sharp drop in MFC performance. To overcome this drawback, metal oxides have been anchored on conducting polymer substrates [99].

Biochar (BC) modified with the nickel ferrite (NiFe_2_O_4_) nanorod/poly(3,4-ethylenedioxythiophene) (PEDOT) composite has been employed as an anode catalyst in MFC [100]. Biochar has been prepared from neem wood carbonization. Electrocatalytic activity, electrochemical impedance spectroscopy (EIS) (Figure 5c), MFC performance (Figure 5a,b), and stability(Figure 5d) of bare BC, Fe_3_O_4_/BC, NiFe_2_O_4_/BC, and PEDOT/NiFe_2_O_4_(1:2)/BC were studied and explained. The voltammograms using the PEDOT/NiFe_2_O_4_/BC catalyst showed superior oxidation current, which confirms the improvement of the bioelectrochemical activity of the exoelectrogens. MFC with PEDOT/NiFe_2_O_4_ (1:2)/BC exhibited the lowest charge transfer resistance (R_ct_) of 441.4 Ω, among all the assembled MFCs (Figure 5c). The coordination bond between the spinel oxide NiFe_2_O_4_ and the PEDOT layer enhanced the electron transfer by lowering the electrode–electrolyte interfacial resistance. Finally, the MFC showed a peak power density of 1200 mWm^−2^ (Figure 5b) mainly because of the unique nanostructure of the catalyst, better electron conductivity, and chemical stability [100].The polyaniline (PANI)/mesoporous TiO_2_ composite was synthesized to be utilized as an anode in MFCs. The composite with 30 wt% PANI exhibited the best catalytic activity by showing a maximum power density of 1495 mW m^−2^, which is found be one of the highest power outputs among all the reported works to date. Large specific surface areas and uniform nanopore distribution of the newly developed nanohybrid could be the key factors of its improved performance [101].

### 3.2. CPNH-Based Cathode for MFCs

One of the major barriers which hinder practical application of the MFC is the slow oxygen reduction reaction (ORR) kinetics and the high cost of the ORR catalysts based on Pt. As a result, the total cost of MFC was enhanced to very high levels along with the lower efficiency. Nowadays, nanostructured carbon such as carbon nanofibers (CNF) [102], carbon nanotubes [103], and activated carbon showed their potential to replace Pt either partially or in full. Non-platinum group metal catalysts based on Mn, Fe, Co, and Cu etc. have also been tested as cathode catalysts for MFCs [104].

Polyindole (PID) and iron phthalocyanine (FePc) was deposited on carbon nanotubes and on vulcan carbon separately. Among all the studied catalysts, FePc/CNTs and FePc/PID/CNTs catalysts exhibited better electrocatalytic activity in terms of more positive half-wave potential values and higher kinetic current density values compared to that of the conventional Pt/C. Finally, the MFC using FePc/PID/CNT cathodes showed a maximum power density of 799 ± 41 mW m^−2^ and a current density of 3480 ± 83 mA m^−2^, which are higher than those achieved with commercial Pt/C cathode (646 ± 25 mW m^−2^ and 3011 ± 84 mA m^−2^, respectively) (Figure 6a). Again, the MFC with FePc/PID/CNT cathodes achieved a stable performance (Figure 6b), which also confirms its potential to replace Pt as cathode catalysts [105].

Iron(II) phthalocyanine (FePc) and cobalt tetramethoxyphenylporphyrin (CoTMPP)-based material had also been synthesized and tested as cathode catalysts for a two-chamber microbial fuel cell. The CoTMPP-based catalyst achieved better maximum current (16.67 mA) and maximum power output (14.32 mW L^−1^) compared to the FePc-based cathode (maximum current 14.31 mA and maximum power output 13.88 mW L^−1^).Stronger back bonding between cobalt and oxygen could be responsible for its better performance. These two inexpensive catalysts also showed comparable electrocatalytic activity to Pt black itself [106]. The manganese-polypyrrole-carbon nanotube (Mn-PPy-CNT) composite was synthesized and used as a cathode catalyst in an air-cathode MFC. Mn-PPy-CNT-based MFC achieved a maximum power density of 169 mW m^−2^ and 213 mW m^−2^ at loadings of 1 mg cm^−2^ and 2 mg cm^−2^, respectively, which are comparable to the MFCs with the commercial Pt/C catalyst. The presence of the Mn–N active site makes this low-cost catalyst potentially applicable as a cathode of the MFCs. Additionally, Mn-PPy-CNT-based MFCs exhibited long-term stability, which is another important factor for practical application of the MFC [107].

Pyrrole was polymerized on reduced graphene oxide (rGO) to use as a catalystsupport. Further, Ni–NiO NPs were deposited on the synthesized support matrix, which was tested as cathode catalysts in single-chambered MFCs. MFCs employing the Ni–NiO/PPy–rGO nanohybrid catalyst exhibited a maximum current density of 2134.56 mA m^−2^ and a maximum power density of ~678.79 ± 34 mW m^−2^, whereas the MFC using the commercial Pt/C catalyst showed lower current and power density, i.e., 1788.2 mA m^−2^ and ~481.02 ± 24 mW m^−2^, respectively. Higher electrical conductivity of the PPy–rGO support matrix provided a good platform for the deposition of the Ni–NiO NPs. The longer conjugation length of PPy–rGO helps to improve the synergistic effect between the metal NPs and the support. Thus, the resulting nanohybrid catalysts led to better electrocatalytic activity and long-term stability compared to the conventional Pt/C cathode [108]. Iron phthalocyanine (FePc) supported on Polyaniline/carbon black (PANI/C) was synthesized and utilized as a cathode catalyst in an air–cathode microbial fuel cell (MFC). PANI/C/FePc achieved better electrocatalytic activity in terms of the ORR peak shift toward positive potential and higher peak current. MFC with the PANI/C/FePc cathode exhibited a maximum power density of 630.5 mW m^−2^, which was found to be higher than the MFC with Pt cathode (575.6 mW m^−2^). Lower cost and higher power output make the PANI/C/FePc catalyst a promising candidate towards an alternative cathode catalyst [109]. The fibrousPani–MnO_2_ nanocomposite was synthesized to investigate its electrocatalytic activity towards ORR. Furthermore, when the PANI–MnO_2_ nanohybrid was used as a cathode catalyst in a MFC that exhibited an improved power density of 0.0588 W m^−2^, the synergistic effect of PANI and MnO_2_ in the nanohybrid catalyst mainly enhanced the contact between the electrode and the electrolyte. As a result, the electronic conductivity and the surface area of the catalyst were improved [110].

## 4. Conclusions

Although many review articles are available in the field of fuel cells based on the design, catalyst, support matrix, and membranes used so far, this review article isparticularlyfocused on conducting polymer nanohybrids materials used in PEMFCs. Here, we have focused on the utilization of CPNHs as catalysts and support matrices in direct methanol fuel cells, direct ethanol fuel cells, and microbial fuel cells. As conducting polymers have unique features in terms of high electronic conductivity due to the presence of the conjugated backbone, better electron delocalization from the CP to the hybrid metal, and high surface area, these properties make the CPNH an attractive material to study their physicochemical properties and to further their application to different types of PEMFCs [111]. Up until now, limited alloy-based multi-metallic efficient electrocatalyst-supported CPs have been studied [112], and in fact, transition metal-based CPNHs have not been reported so far. Such low costs may improve the catalytic efficiencies and performance of fuel cell energy devices.

## Figures and Tables

**Figure 1 polymers-12-02993-f001:**
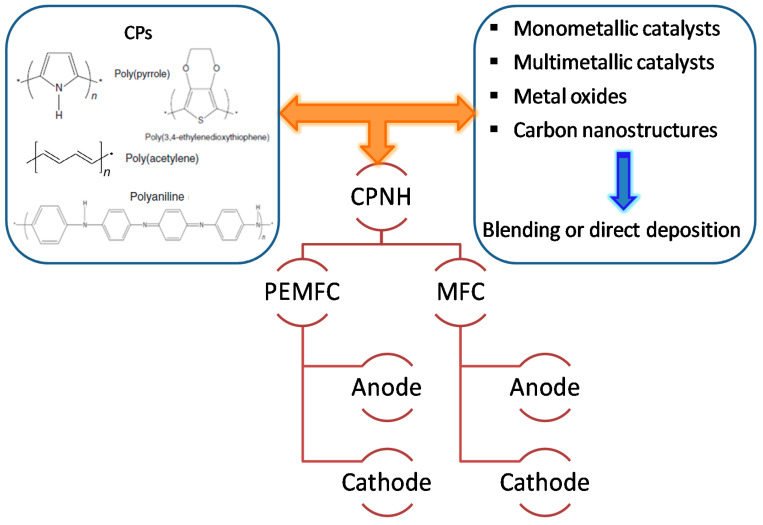
The active components of conducting polymer-based nanohybrids (CPNHs) in fuel cell materials.

**Figure 2 polymers-12-02993-f002:**
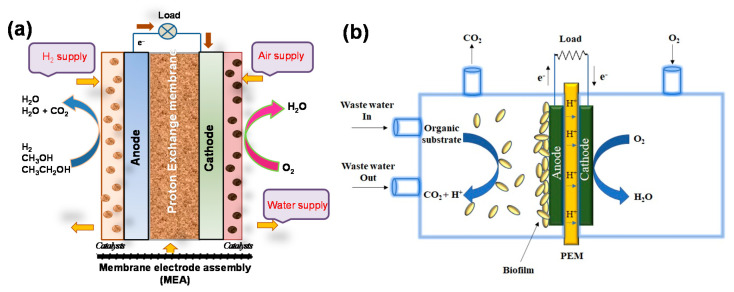
Schematic illustration of the working principle in (**a**) direct alcohol fuel cells and (**b**) microbial fuel cells.

**Figure 3 polymers-12-02993-f003:**
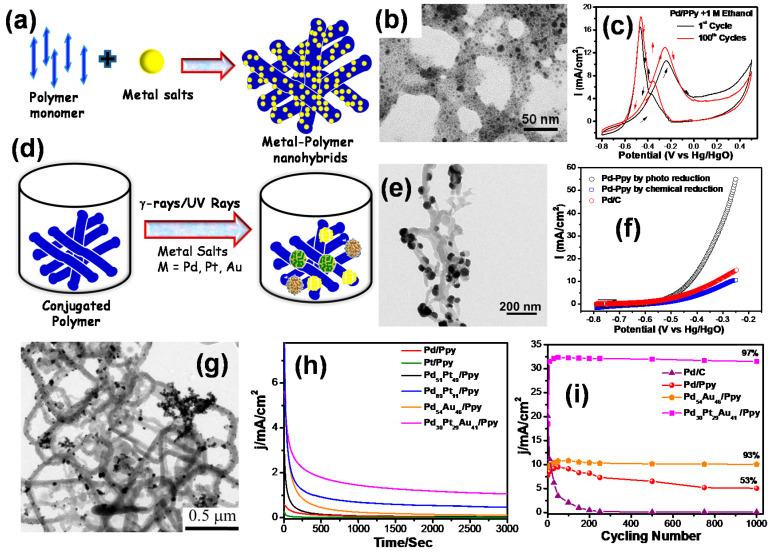
(**a**) Schematic representation of in situ metal–polymer nanohybrids; (**b**) transmission electron micrograph of Pd/PPy NHs;(**c**) cyclic voltammograms of Pd/PPy NHs and superposition of the first (black solid line curve) and the 100th (red solid line curve) scans for the electrocatalytic oxidation of 1 M EtOH in 1 KOH; (**d**) formation of metal nanoparticles on conducting polymer nanofibers by radiolysis or UV radiation; (**e**) transmission electron micrograph of the Pd/PPynanohybrid; (**f**) the forward scan peaks of the cyclic voltammetry (CV) of Pd/PPy via photoreduction, of Pd/PPyvia chemical reduction, and of Pd/C for the electrocatalytic oxidation of 1 M EtOH [49]; (**g**) typical TEM images of Pd_30_Pt_29_Au_41_/PPy nanohybrids; (**h**) chronoamperometric curves for ethanol oxidation at constant potential −0.25 V vs. Hg/HgO on Pd/PPy, Pt/PPy, Pd_89_Pt_11_/PPy, Pt_49_Pd_51_/PPy, Pd_54_Au_46_/PPy, and Pd_30_Pt_29_Au_41_/PPy electrodes; and (**i**) long cycling study of Pd/C, Pd/PPy, Pd_54_Au_46_/PPy, and Pd_30_Pt_29_Au_41_/PPy electrodes in a solution of 1M KOH and 1M ethanol at a scan rate of 50 mV s^−1^ [50].

**Figure 4 polymers-12-02993-f004:**
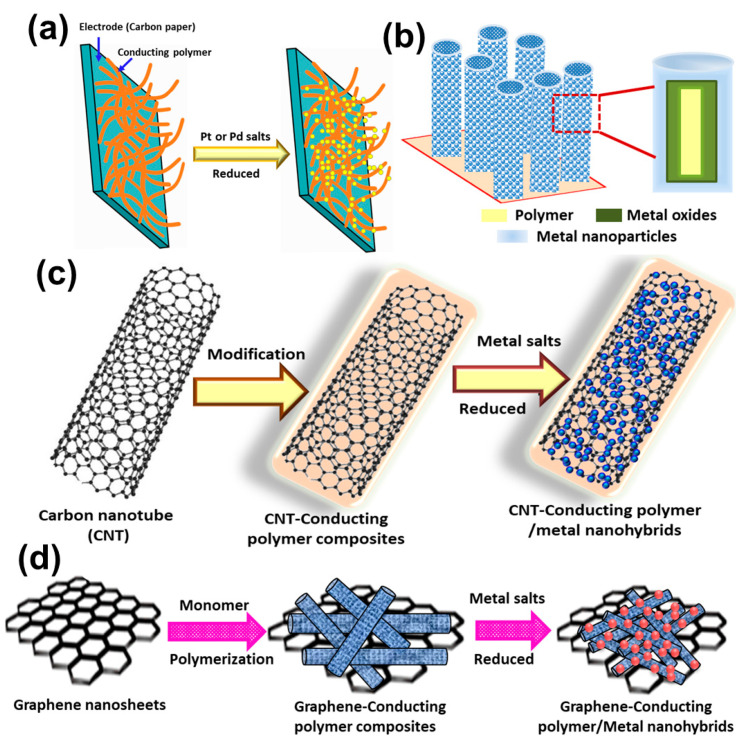
Schematic representations of polymer-modified support-based nanohybrid electrodes: (**a**) carbon electrode–polymer/metal, (**b**) metal oxides–polymer/metal, (**c**) carbon nanotubes–polymer/metal, and (**d**) graphene–polymer/metal.

**Figure 5 polymers-12-02993-f005:**
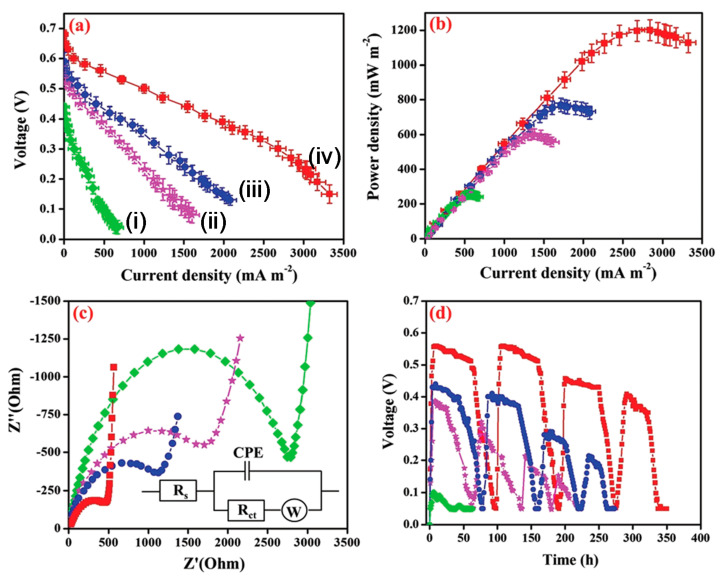
(**a**) Polarization and (**b**) power density curves (error bars indicate the standard deviation acquired from three microbial fuel cell (MFC) experiments), (**c**) electrochemical impedance spectroscopy (EIS) analysis, Inset: depicting the equivalent circuit used for fitting the EIS data, and (**d**) durability performances of MFCs with (green dots) bare Biochar (BC) anode (i), (violet dots) Fe_3_O_4_/BC (ii), (blue dots) NiFe_2_O_4_/BC (iii), and (red dots) PEDOT/NiFe_2_O_4_(1:2)/BC (iv) [101].

**Figure 6 polymers-12-02993-f006:**
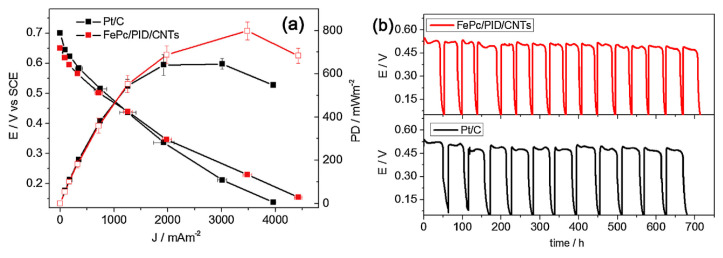
(**a**) Cell voltage and power density of MFCs assembled with FePc/PID/CNTs and Pt/C cathodes, and (**b**) voltage cycles under a 1 kV external resistance by MFCs assembled with FePc/PID/CNTs_CC and Pt/C_CC cathodes, recorded for 700 h after inoculation [105].

**Table 1 polymers-12-02993-t001:** Conducting polymer-based nanohybrid materials and their applications in electrocatalysis.

Electrode	Synthesis Method	Fuel Cell Reaction	E_onset_ mV	I_f_, mA.cm^−2^	Reference
Pd/PPy	Chemical polymerization	EOR	−628	7.05	[47]
Pd/PEDOT	Electrochemical method	EOR	−562	137	[48]
Pd/PPy	Chemical polymerization followed by photo reduction	EOR	−708	53.8	[49]
Pd/PPy	Chemical polymerization followed by radiolysis	EOR	−640	9.50	[50]
Pd_30_Pt_29_Au_41_/PPy	Chemical polymerization followed by radiolysis	EOR	−630	32.45	[50]
Pd/PPy	‘water-in-oil’ microemulsion for chemical polymerization	ORR	-	-	[53]
Au/PANI	Interfacial chemical polymerization	ORR	−100	-	[54]
Au-V_2_O_5_/Polyindole	Wet chemical method followed by chemical polymerization		−400	-	[55]
Pt/PPy nanofibers	Interfacial polymerization	MOR	-	14.1	[62]
Pt/PPy nanofibers	Iinterfacial polymerization	MOR	-	-	[63]
Pt nanocube assemblies/PANI	Wet-chemical approach followed by chemical polymerization	MOR	-	0.85	[64]
Ni/SPAni	Chemical polymerization followed by chemical reducing agent	MOR	-	2.15	[65]
PtPd/PPy/PtPd	Electrochemical synthesis via galvanostatic electrodeposition	MOR	250	0.9	[51]
Pt–Fe/PPy	In situ interfacial polymerization	MOR	170	-	[60]
Pt_66_Pd_34_/PPy	Chemical polymerization followed by radiolysis	MOR	222	8.14	[70]
Pt_24_Pd_26_Au_50_/PPy	Chemical polymerization followed by radiolysis	MOR	227	6.76	[70]

**Table 2 polymers-12-02993-t002:** List of the functionalized conducting polymer-supportedmetal electrocatalysts for fuel cell applications.

Electrode	Functional Unit as Support	Fuel Cells Reaction	Ref.
Pt/C/PEDOT	Carbon paper coated 3,4-polyethylenedioxythiophene	MOR	[72]
Pd nanodendritic /C/PEDOT	Carbon paper coated 3,4-polyethylenedioxythiophene	Alcohols oxidation	[73]
Pt nanocactus /PANI/ CFP	Poly (aniline) decorated with platinum on carbon fiber paper	Toluene oxidation	[74]
Pd nanoplates /PDPB/Nafion	Nafion modified poly(diphenylbutadyine) nanofiber	EOR	[75]
Pt/C-PANI	Vulcan XC-72 and PANI-doped with trifluoromethane sulfonic acid	MOR	[76]
Pt/PEDOT/PSS	3,4-polyethylenedioxythiophene modified with poly(styrene sulfonic acid) (PSS)	MOR	[77]
Pt-Ru/PANI/PSS	Polyaniline–poly(styrene sulfonic acid) (PSS)	MOR	[78]
Pt-PFSA/C/PANI	Perfluorosulfonic acid, PFSA and polyaniline	ORR	[79]
Pt/PPy-C	Polypyrrole-carbon	borohydride oxidation and hydrogen peroxide reduction	[80]
Pt/H_x_MoO_3_/PEDOT-PSS	Pt NPs and H_x_MoO_3_deposited in poly(3,4-ethylenedioxythiophene)-poly(styrene sulfonic acid)	MOR	[81]
PANI-PAMA-Pt-RuO_2_	Polyaniline doped with poly(acrylic acid-co-maleic acid) (PAMA) and electrodeposition of RuO_2_	MOR	[82]
Pt/CeO_2_/PANI	Polyaniline with CeO_2_ as multilayered supporting material	MOR	[83]
ZnO/Pt/CeO_2_/PANI	Polyaniline with ZnO-CeO_2_ as multilayered supporting material	MOR	[83]
Pt–Ru/PPy–CNT	Polypyrrole/multiwalled carbon nanotubes	MOR	[87]
Pt–Ru/PTh-CNTs	Polythiophene/CNT composites (PTh-CNTs)	Ethylene glycol oxidation	[88]
Cobalt-PPy/MWCNT	Polypyrrole-multiwalled carbon nanotube	ORR	[89]
Pt/MnOx–PEDOT–MWCNTs	Manganese oxide-poly(3,4-ethylenedioxythiophene)-carbon nanotubes composite	MOR	[90]
Pd nanoflowers /PPy/MWCNTs	Polypyrrole-multiwalled carbon nanotube	MOR	[91]
Pd /PANI/GNS	polyaniline-reduced graphene oxide hybrid nanosheets	MOR and EOR	[93]
Pd/PEDOT/graphene	Poly(3,4-ethylenedioxythiophene) nanosphere-graphene nanosheets	EOR	[94]
Pd/PEDOT/rGO	Poly(3,4-ethylenedioxythiophene) functionalized graphene	ORR	[95]
Pd/graphitic carbon nitride/PANI	Graphitic carbon nitride-polyaniline	MOR	[96]

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
