# Peer review of "Conducting Polymer-Based Nanohybrids for Fuel Cell Application"

_polymers, 2020, doi:10.3390/polym12122993_

Round 1

Reviewer 1 Report

The review describes the conducting polymer-based nanohybrids for fuel cells application. 

In the abstract:

Conducting polymers are not strictly carbon materials. For example, polypyrrole is based on C and N. Please improve.

The abstract do not include any quantitative data. Please add some details in this regard.

The figure 2 is need the copyrights permission?

The same situation for all the figures.

Some details regarding the electrochemical deposition of conducting polymers must be included.

The characterization of the catalysts must be included.

Author Response

Reviewer 1

The review describes the conducting polymer-based nanohybrids for fuel cells application. 

In the abstract:

Conducting polymers are not strictly carbon materials. For example, polypyrrole is based on C and N. Please improve.

Thank you for this comment. Necessary corrections have been made in the revised manuscript.

The abstract do not include any quantitative data. Please add some details in this regard.

We agree with the reviewer. Thank you for the suggestion.

The figure 2 is need the copyrights permission? The same situation for all the figures.

We agree with the reviewer. We have collected copyrights permission files for all the figures.

Some details regarding the electrochemical deposition of conducting polymers must be included.

Thank you for pointing out this issue. We made correction in the revised version of the manuscript. 

The characterization of the catalysts must be included.

Thank you for this suggestion. In the revised manuscript, we discuss in details the characterization of the catalysts as reviewer suggested.

Reviewer 2 Report

After checking the topic and progress in conducting polymer nano hybrids for fuel
cells, I have found a recent publication that the authors missed during their literature survey that deal with the same subject: A review of progressive advanced polymer nanohybrid membrane in fuel cell application by Zakarias et al. Int J Energy Res. 2020;1–41.doi: 10.1002/er.5516. Furthermore, they also missed other important publication on the subject: Han et al. Conducting Polymer-Noble Metal Nanoparticle Hybrids: Synthesis Mechanism Application. Progress in Polymer Science, 2017, http://dx.doi.org/doi:10.1016/j.progpolymsci.2017.04.002.

Therefore, I regret to inform to you that your manuscript has to be rejected because of the lack of novelty.

Author Response

Reviewer 2

Comments and Suggestions for Authors

After checking the topic and progress in conducting polymer nano hybrids for fuel cells, I have found a recent publication that the authors missed during their literature survey that deal with the same subject: A review of progressive advanced polymer nanohybrid membrane in fuel cell application by. Zakarias et al. Int J Energy Res. 2020;1–41.doi: 101002/er.5516. Furthermore, they also missed other important publication on the subject: Han et al. Conducting Polymer-Noble Metal Nanoparticle Hybrids: Synthesis Mechanism Application. Progress in Polymer Science, 2017, http://dx.doi.org/doi:10.1016/j.progpolymsci.2017.04.002.

Therefore, I regret to inform to you that your manuscript has to be rejected because of the lack of novelty.

Thank you for this comment and suggested references. We have included the suggested references in the revised manuscript.

Zakarias et al. highlights fabrication and functionalization of nanohybrid membrane composed of inorganic fillers and organic polymer like Nafion for fuel cell applications. Conjugated polymer nanostructures are not included in the particular review. In another review, Han et al. cover strategies for the synthesis of CP-based hybrids, and CP noble metal nanoparticle hybrids and its potential applications in the fields of catalysis, sensor, surface-enhanced Roman scattering (SERS) device and others. In fact, electrochemical oxidation of organic molecules is not included in the catalysis part of the review article. Hence both of the proposed review articles content is not overlapping with the present manuscript.

The concept of conducting polymer materials emerged in recent decades and continues to attract scientific community. Many excellent reviews of the knowledge accumulated regarding the development of conducting polymer and their applications in the field such as sensing and device have been appeared. However, preparation, characterization and application of conducting polymer nanohybrids in fuel cell applications are still at the foreground of research activity particularly in energy conversion application. There is a need for deep understanding of the availability of new polymer nanohybrids. The present review includes synthesis of polymer nanohybrid and the evaluation of their effectiveness in fuel cell application.

Reviewer 3 Report

The authors have presented an interesting review article summarizing information on fuel cells. At the same time, it is strongly recommended that certain aspects be paid intensive attention to. In particular, the text is far from being reader-friendly and thus has to be recomposed. Spelling and grammar must be checked attentively to eliminate mistakes like that made in the sentence “Typically, electron flows from the anode to the cathode through an external electrical connection” whereas it is expected to contain the phrase “electrons flow”. Yet, it is necessary to stick to a single standard of writing formulae in order to avoid variations like PPY vs. Ppy or platinum-PANI vs. Pt/C. Also, it will help a lot if such terms as “CoTMPP based catalyst” are delivered in the hyphenated form like “CoTMPP-based catalyst”. Also, expand the abbreviation PoPD.

The explanatory phrases contained in Figure 1 are somewhat confusing. In particular, the word “polypyrrole” has to be placed next to “PPy”, the same with “PEDOT”. It is doubtable that identifying the polymer with triple bonds using the name “polyacetylene” is appropriate, for polyacetylene contains double bonds only. The presented formula rather corresponds to the name “carbyne”.   Fig. 5 It is desirable to put the numbers for each curve in Figure 5 next to the curve and give the number in the figure caption.
The review is good quality. 

Author Response

Reviewer 3

Comments and Suggestions for Authors

The authors have presented an interesting review article summarizing information on fuel cells. At the same time, it is strongly recommended that certain aspects be paid intensive attention to. In particular, the text is far from being reader-friendly and thus has to be recomposed. Spelling and grammar must be checked attentively to eliminate mistakes like that made in the sentence “Typically, electron flows from the anode to the cathode through an external electrical connection” whereas it is expected to contain the phrase “electrons flow”. Yet, it is necessary to stick to a single standard of writing formulae in order to avoid variations like PPY vs. Ppy or platinum-PANI vs. Pt/C. Also, it will help a lot if such terms as “CoTMPP based catalyst” are delivered in the hyphenated form like “CoTMPP-based catalyst”. Also, expand the abbreviation PoPD.

Thank you for the comment. Necessary corrections have been made in the revised manuscript. We included correction in the revised manuscript.

The explanatory phrases contained in Figure 1 are somewhat confusing. In particular, the word “polypyrrole” has to be placed next to “PPy”, the same with “PEDOT”. It is doubtable that identifying the polymer with triple bonds using the name “polyacetylene” is appropriate, for polyacetylene contains double bonds only. The presented formula rather corresponds to the name “carbyne”.   Fig. 5 It is desirable to put the numbers for each curve in Figure 5 next to the curve and give the number in the figure caption.

We modify the Figure 1 and Figure 5 in the revised manuscript as suggested by the reviewer.

The review is good quality.

Thank you for the comment.

Round 2

Reviewer 1 Report

The manuscript could be published without further modifications. 

Reviewer 2 Report

The authors work hard to show the main differences between their re-submitted paper and the one previously published by Zakarias et al.

They also addressed the rest of my comments in appropriate form. However, I recommend the authors to modify their article Title to emphasize that this is a review article.